# Novel Classification of Thrombotic Disorders Based on Molecular Hemostasis and Thrombogenesis Producing Primary and Secondary Phenotypes of Thrombosis

**DOI:** 10.3390/biomedicines10112706

**Published:** 2022-10-26

**Authors:** Jae Chan Chang

**Affiliations:** Department of Medicine, School of Medicine, University of California Irvine School of Medicine, Irvine, CA 92868, USA; jaec@uci.edu; Tel.: +1-949-943-9988

**Keywords:** thrombosis, microthrombosis, fibrin clot disease, macrothrombosis, combined micro-macrothrombosis, hemostasis, thrombogenesis, microthrombogenesis, fibrinogenesis, macrothrombogenesis, endotheliopathy, vascular microthrombotic disease

## Abstract

Thrombosis, the common and deadliest disorder among human diseases, develops as a result of the intravascular hemostasis following an intravascular injury, which can be caused by a variety of trauma, non-traumatic insults or clinical illnesses. Thrombosis can occur at any location of the vascular system supplied by blood from the heart to large and smallest arterial and venous systems and may affect the function and anatomy of the organ and tissue. It more commonly occurs in the smaller circulatory system of the vascular tree such as arterioles and capillaries, and venules of the organs, especially in the brain, lungs, heart, pancreas, muscle and kidneys, and sinusoids of the liver. Thrombosis has been referred as the disease of “blood clots”, which concept is incompletely defined, but represents many different hemostatic diseases from microthrombosis to fibrin clot disease, macrothrombosis, and combined micro-macrothrombosis. Thrombosis is produced following an intravascular injury via one or more combination of four different mechanisms of thrombogenesis: microthrombogenesis, fibrinogenesis, macrothrombogenesis and micro-macrothrombogenesis initiated by normal physiological hemostasis in vivo. The clinical phenotype expression of thrombosis is determined by: (1) depth of the intravascular wall injury, (2) extent of the injury affecting the vascular tree system, (3) physiological character of the involved vascular system, (4) locality of the vascular injury, and (5) underlying non-hemostatic conditions interacting with hemostasis. Recent acquisition of “two-path unifying theory” of hemostasis and “two-activation theory of the endothelium” has opened a new frontier in science of medicine by identifying the pathophysiological mechanism of different thrombotic disorders and also contributing to the better understanding of many poorly defined human diseases, including different phenotypes of stroke and cardiovascular disease, trauma, sepsis and septic shock, multiorgan dysfunction syndrome, and autoimmune disease, and others. Reviewed are the fundamentals in hemostasis, thrombogenesis and thrombosis based on hemostatic theories, and proposed is a novel classification of thrombotic disorders.

## 1. Introduction

From the ancient Greek civilization in human history, the mechanism of hemostasis has been thought as an intriguing science of medicine in bleeding patient with the vascular injury [1]. To date, the basic framework of hemostasis is still partially understood, but clinicians have attempted the interpretation for a hemorrhagic event and thrombosis based on contemporary theory of tissue factor (TF)-FVIIa complex-initiated extrinsic cascade activation more than a half century. It has allowed, but incompletely, to define the diagnosis of hemostatic disorders in hemorrhage and thrombosis and modify their treatments with each step of advancing knowledge of coagulation science [2]. However, the poorly understood mechanism of thrombogenesis has masked the pathogenesis of many human diseases when hemostatic nature plays the major role leading to thrombosis that produces metabolic dysfunction due to altered blood circulation and supply of oxygen, and molecular and anatomical changes for the cells, tissues and organs.

In recent decades, the fundamental concept of hemostasis (cessation process of hemorrhage) and coagulation (forming process of thrombosis) has been understood to be the same or similar mechanism although the former is physiological mechanism and the latter results in a pathologic disorder [3]. This basic concept has not been embraced easily without skepticism because the term hemostasis carries two paradoxically different connotations in the comprehension of hemorrhage and thrombosis [3,4]; first, it is an essential system protecting human lives in the bleeding patient occurring in external and internal bodily injury, and second, it is also the pathologic disorder harming human lives by forming deadly blood clots within the vessel in intravascular injury. In the former, hemostasis would promote the wound healing to save lives, but, in the latter, it could produce a serious thrombotic disease jeopardizing lives [4]. It is an irony that nature endowed human with one hemostatic mechanism that repairs a life-threatening wound, but also produces a deadly thrombotic disorder.

In this essay article, I will briefly review recently identified novel hemostatic theories and elaborate the mechanism of thrombogenesis to show how hemostasis contributes to the complexity of thrombotic diseases in intravascular injury and attempt to identify the variety of clinical phenotypes of thrombosis centered on two hemostatic theories: “two-path unifying theory of hemostasis” [3,4] and “two-activation theory of the endothelium” [5] schematically illustrated in Figure 1 and Figure 2. Further, a molecular-based pathogenetic classification of thrombotic disorders will be formulated.

## 2. Retrospective on Hemostasis and Thrombosis

The contemporary theory of hemostasis has been conceptualized for the care of the patient to assist the diagnosis and treatment for hemorrhagic disease in external or internal bodily injury and thrombosis in intravascular injury [6]. The fundamental of hemostasis has been derived from the theory that the tissue factor (TF)-initiated blood coagulation activating FVII and producing fibrin clots plays the key role. The TF-FVIIa complex in TF pathway of in vivo hemostasis promotes sequential activation of coagulation proteins and serine proteases via the extrinsic clotting cascade. This TF pathway activates FIX to FIXa, which activates FX to FXa leading to FV activation to form FXa-FVa complex formation (prothrombinase) [7] that converts prothrombin to thrombin and facilitates fibrinogen to fibrin clot formation. In external bleeding injury (e.g., trauma) the formed blood clots made of microthrombi strings and fibrin clots become “hemostatic plug”, but in intravascular injury (e.g., surgery) the blood clots become “thrombus”, which disorder is called “thrombosis” [4].

However, the concept that the “fibrin clots” formed via the mechanism of fibrinogenesis (i.e., TF path of coagulation) has not been congruous with the fundamental notion of “hemostasis”, “thrombogenesis” and “thrombosis”. First, the components of ultra large von Willebrand factor (ULVWF)/FVIII and platelets could not be reconciled within fibrin clots formed by fibrinogenesis promoted by the extrinsic clotting cascade in the bleeding patients. Second, because of incomplete model of contemporary hemostasis in vivo (i.e., TF path) for the role of the platelet and factor VIII and FIX in vitro coagulation test system, the complete picture of the genesis of thrombosis could not be explained by molecular thrombogenetic mechanism [4]. Finally, the traditional hypothesis of Virchow’s triad, consisting of (1) stasis of blood flow, (2) endothelial injury, and (3) hypercoagulability, introduced as the primary event and accepted to be the essential components producing “thrombosis” in intravascular space contradicts the concept of vascular injury triggering platelet activation and releasing ULVWF from the vascular wall to initiate hemostasis and thrombosis. The rational mechanism of thrombogenesis has not been identified by molecular hemostasis. [8,9]. The unidentified pathophysiological process of “thrombogenesis” due to the dogmatic approach of Virchow’s triad has buried the true mechanism of hemostasis and thrombosis in the background of the medical science more than one century to date.

It is my belief that this erroneous conceptualization on thrombogenesis has delayed identifying the true mechanism of hemostasis in vivo due to misinterpretation on the terms amongst “hemostasis”, “coagulation”, and “thrombosis”. I have attempted to redefine them in Table 1 to find their true meanings [4]. In retrospect, this incorrectly understood relationship between hemostasis and thrombosis has impeded not only discovering the physiological hemostatic mechanism, but also precluded identifying the pathological mechanism of thrombogenesis and determining the true character of many expressive thrombotic diseases, including “disseminated intravascular coagulation” (“DIC”), coagulopathy, hypercoagulable state, microthrombosis, deep venous thrombosis, and organ dysfunction syndromes. For the same reason, the damaged vascular repair mechanism initiated by the platelet and ULVWF has not been supported by the essential components of in vivo hemostatic process initiating “thrombus” formation along with “fibrin clots” via TF-FVIIa complex-initiated extrinsic clotting cascade (i.e., fibrinogenesis) [3].

## 3. The Identity of Thrombosis

Since “two-path unifying theory” of hemostasis was published, it has been affirmed that hemostasis is blood clotting mechanism forming “hemostatic plug” in bleeding from external and internal bodily injury, and is also thrombosis promoting mechanism in intravascular injury by producing “thrombus” [3,4]. Succinctly speaking, hemostasis is the tool to stop the bleeding in bodily injury, but thrombosis is the harmful product made by the tool in intravascular injury. Based on this comprehension of the concept in hemostasis and thrombosis, three fundamentals in normal and abnormal hemostasis are summarized in Table 2.

### 3.1. Vascular Injury and Hemostasis

The most important principle is “Hemostasis can be activated only by vascular injury”. As shown in Figure 3 on the schematic histological structure of vascular wall, it is critically important to understand the damaged blood vessel wall must be repaired by the released hemostatic components (i.e., ULVWF/FVIII and TF) from the vessel wall in cooperation with coagulation factors present in blood (i.e., platelet, FVII, FIX, FX, FV, FII and fibrinogen) at the damaged area of the vessel wall. This is true in both external and internal bodily injury, and in intravascular injury [3]. Without a vascular injury, there is no release of ULVWF/FVIII from the endothelium (i.e., ECs) and TF from the subendothelial tissue (SET) and/or extravascular tissue (EVT), which means hemostasis never occurs if there is no release of ULVWF/FVIII and/or TF. In another word, hemostasis is not needed without a vascular damage, and thrombosis/coagulation does not develop in the absence of vascular injury. For example, a hemophilic person with FVIII deficiency never bleeds in the absence of a vascular injury. So is very true that thrombosis and coagulopathy do not occur without a vascular injury. This concept has a very important philosophical impact in the management of every hemostatic disorder.

The second principle is “Hemostasis must be activated through ULVWF path and/or TF path”, which differentiate microthrombosis (i.e., endotheliopathy) due to activated ULVWF path in vascular wall damage limited to ECs from macrothrombosis (e.g., deep venous thrombosis) due to activated ULVWF path and activated TF path in vascular damage extending from EC to SET/EVT. Microthrombosis is the result of microthrombogenesis, and macrothrombosis is the result of macrothrombogenesis as depicted in Figure 1. This notion supports that activated ULVWF path in extensive endotheliopathy produces microthrombi strings composed of platelet-ULVWF/FVIII complexes, but activated TF path with minimum release of ULVWF in localized intravascular trauma produces macrothrombus made of platelet-ULVWF/FVIII complexes and fibrin-thrombin clots. This principle supports (1) vascular injury is the initiating point of hemostasis, (2) every thrombosis except a few conditions—which will be discussed later—should be the result of vascular injury, and (3) bleeding or thrombosis does not occur without a vascular injury, even in hemophilia and thrombophilia [4].

The primary clinical phenotype of thrombosis is determined by the depth, extent and location of the vascular injury and intrinsic character of thrombus formed at different vascular milieux. Further, endotheliopathy is typically associated with ECs damage occurring in the disease such as sepsis (e.g., bacterial and viral) resulting in disseminated endotheliopathy-associated vascular microthrombotic disease (e.g., EA-VMTD), but ECs/SET/EVT damage occurs with a local vascular injury such as trauma and in-hospital vascular accesses (e.g., surgery, indwelling vascular device) resulting in local macrothrombus. The former is a systemic disease due to activated ULVWF path and produces disseminated microthrombosis, but the latter is a localized disease due to activated TF path with minimum ULVWF path and produces local macrothrombosis.

Therefore, three thrombogenetic mechanisms can be recognized in a vascular injury depending upon the depth of vascular wall damage and the extent of vascular tree involvement. These are (1) microthrombogenesis, (2) fibrinogenesis, and (3) macrothrombogenesis as illustrated in Figure 1. This basic conceptual understanding of hemostasis has major implication in identifying and classifying the phenotypes of thrombotic disorders in intravascular injury.

### 3.2. Intravascular Injury, Thrombosis and Thrombogenesis

The intravascular injury involving ECs and/or SET/EVT from the inside of the vessel wall provokes endotheliopathy and/or intravascular bleeding, which triggers intravascular hemostasis leading to microthrombogenesis and macrothrombogenesis [3,4]. Thus, “thrombogenesis” is a term denoted for “activated hemostasis producing bloods clots in intravascular injury”. It includes microthrombogenesis producing microthrombi strings, fibrinogenesis producing fibrin meshes, and finally macrothrombogenesis forming macrothrombus by the unifying mechanism of microthrombi strings and fibrin meshes (Figure 1). Until recently, molecular pathogenesis of thrombosis has not been established despite of intensive theoretical studies to prove the dogmatic concept of Virchow’s triad. The term “thrombosis” broadly encompasses every disease produced by every blood clot, caused by the disorders due to fibrin clots, which unknowingly has included microthrombi, macrothrombus, and combined micro-macrothrombi present in the arterial and venous systems. Additionally, included are non-hemostatic thrombotic diseases that are not triggered by a vascular injury. They are thrombotic thrombocytopenic purpura (TTP), fibrin clot disease of acute promyelocytic leukemia (APL), and heparin-induced thrombocytopenia with white clot syndrome (HIT-WCS). These three disorders are not due to hemostatic activation since they occur without a vascular injury [4].

### 3.3. Thrombosis as Generic Term

The descriptive terms for thrombosis such as deep venous thrombosis (DVT), aortic arch thrombosis, acute ischemic stroke (AIS), acute myocardial infarction (AMI), venous thromboembolism (VTE), pulmonary thromboembolism (PTE), inferior vena cava thrombosis (IVCT), superior vena cava thrombosis (SVCT), splanchnic vein thrombosis (SVT), portal vein thrombosis (PVT), Budd-Chiari syndrome (BCS), cerebral venous sinus thrombosis (CVST), microvascular thrombosis, thrombotic thrombocytopenic purpura (TTP), TTP-like syndrome, sinusoidal obstructive syndrome (SOS), veno-occlusive disease (VOD), symmetrical peripheral gangrene (SPG), diabetic gangrene, thrombosis of paroxysmal nocturnal hemoglobinuria (PNH), vaccine-associated thrombocytopenia and thrombosis, and disseminated intravascular coagulation (“DIC”) have been customarily utilized to express localization-denoted diagnosis of thrombotic disorders. However, their thrombogeneses based on underlying molecular events have not been established in any of each thrombosis, but presumed to be the results of activated TF path-initiated coagulation cascade following vascular injury. Thus, the prevailing concept has been all of the thrombosis is the same disease in their character due to the same hemostatic (coagulation) process, and differently expressed phenotypes are the results of different vascular localization (e.g., artery/vein, macrovasculature/microvasculature, or vessels of the heart, lungs, brain, kidneys and other organs).

Now it is obvious that “thrombosis” is a non-specific generic term representing “blood clot(s)” without defining its origin, pathogenesis, character, pathology and/or clinical expression of dysfunction of the organ or tissue. This term provides an insufficient description in applying to clinical practice and educational system for the diagnosis and treatment [10,11]. Further, thrombosis has been improperly thought to be blood clot(s) caused by risk factors such as old age, affluent economic status, immobilization, prolonged high altitude flying, drug, pregnancy, and associated diseases such as diabetes, infection, autoimmune disease, cancer and sepsis, thrombophilia, and even earthquake. To the contrary, according to hemostatic fundamentals (Table 2), the true risk “event” of thrombosis is only “one”, which is an intravascular wall damage resulting from intravascular injury. Indeed, the above observational risk factors are not true risk factors for thrombosis, but are rather contributing events to vascular injury or potentiating factors for thrombosis after a vascular injury has occurred [12].

The use of the simplified term “thrombosis” to define every “blood clot(s)” disorder should be discouraged and replaced with better term representing its genesis, pathogenesis, character, and/or pathology with consideration of localization in the clinical medicine and scientific medical literature. For example, “microthrombosis” in the lungs of COVID-19 sepsis is a well-defined hemostatic disorder provoking acute respiratory distress syndrome (ARDS) [13] caused by EA-VMTD. Additionally, “macrothrombosis” of PTE in the lungs travelled from VTE of the venous system is a well-defined hemostatic disorder provoked by another vascular injury (e.g., vascular access), which vascular injury is not directly related to ARDS [14]. These two disorders, microthrombosis of ARDS and macrothrombosis from PTE in COVID-19 infection, are uniquely different in their genesis, pathogenesis and pathology as well as the intrinsic character of thrombi [11,14]. Sometimes other “thrombosis” phenotypes occurring COVID-19 infection or its vaccination have included not only microthrombosis and macrothrombosis, but also other descriptive terms such as CVST [15,16], SVT [17], SPG [18], acrocyanosis [19], COVID toes [20] and many others. These are neither microthrombosis such as ARDS nor common macrothrombosis seen in distal DVT in their thrombotic character, but are the manifestations of combined micro-macrothrombosis resulting from the unifying mechanism of two different origins of blood clots which are the microthrombi strings from sepsis and the fibrin meshes from in-hospital vascular injury [10,11]. Their pathogenesis will be examined later.

## 4. In Search of New Classification for Thrombotic Disorders

Without the comprehension of pathophysiological mechanism of hemostasis and thrombogenesis, clinical phenotype classification of thrombosis is an impossible task. To be able to estimate the prognosis and recommend proper therapeutic regimens in variety of thrombotic disorders, a novel classification is urgently needed. Fortunately, we have two hemostatic theories as tools to understand the molecular pathogenesis of hemostasis and thrombogenesis in each phenotype of thrombotic disorders. Although hemostasis in intravascular injury initiates and progresses through three thrombogenetic paths (Figure 1) that can be called the blood clotting process to stop bleeding, it may be only the beginning of paths of thrombogeneses before generating the variety of phenotypes. Thrombogenetic mechanism triggers different paths of hemostasis depending upon the specific nature of vascular injury producing a spectrum of thrombotic disorders and is also modified by interaction with other pathologic and pathogenetic factors as follows.

### 4.1. Issues to Be Considered in Classification of Thrombotic Disorders

The failure of identifying the mechanism of thrombogenesis has been primarily due to not focusing on the role of blood vessel wall anatomy, function and circulation physiology in vascular injury and not incorporating them to hemostasis in vivo. Additionally, the underlying cause of vascular injury (e.g., surgery, indwelling vascular catheter), other concurrent disease (e.g., diabetes, cancer) interacting with hemostasis, environmental factor (e.g., sepsis, drug), and modifying genetic component (e.g., ADAMTS13 insufficiency, thrombophilia) are now recognized as important contributors to the complexity of thrombogenesis as detailed in Table 3.

### 4.2. Factors Caused by the Vascular Damage

In intravascular injury, three major factors from the blood vessel damage influence hemostasis and thrombogenesis, which are (1) vascular characteristics of anatomy, function and blood flow (i.e., artery vs. vein) [10,11], (2) depth of vascular wall damage (i.e., ECs vs. ECs/SET/EVT) [3,4], and (3) extent of vascular tree involvement (i.e., local vs. disseminated, microvasculature vs. macrovasculature) [21,22].

The difference between arterial system and venous system are very significant due to their characteristics on oxygen delivery and CO_2_ disposal, direction of blood flow, vascular pressure and shear stress, and attracting neutrophil extracellular traps (NETs) as shown in Table 4. This variance determines the clinical phenotypes between arterial endotheliopathy-associated VMTD (aEA-VMTD) and venous endotheliopathy-associated (vEA-VMTD), and also between arterial thrombosis (e.g., acute ischemic stroke) and venous thrombosis (e.g., DVT) illustrated in Figure 4 [10,11]. The anatomic and physiologic difference between arterial and venous milieux can fabricate remarkably different EA-VMTD. Arterial endotheliopathy often produces the triad of TTP-like syndrome characterized by consumptive thrombocytopenia, MAHA and MODS [23], but venous endotheliopathy typically causes immune thrombocytopenic purpura (ITP)-like syndrome with “silent clinical feature” [10]. Surprisingly, this disparity expressed in microthrombotic syndromes in COVID-19 pandemic has provided me a clue that COVID-19 is predominantly the disease of vEA-VMTD (e.g., ARDS and ITP-like syndrome) [13] while Ebola viral hemorrhagic disease is commonly associated with aEA-VMTD causing hepatocellular necrosis (e.g., hepatic coagulopathy and TTP-like syndrome) [24], which had been thought to be acute “DIC”.

The depth of vascular wall damage (Figure 3) plays the most important role determining the path of hemostasis and triggers one or more thrombogenetic mechanisms of microthrombogenesis and fibrogenesis, later with or without macrothrombogenesis according to the principle summarized in Table 2 and Table 3. The damage limited to ECs produces microthrombosis via microthrombogenesis initiated by lone activation of ULVWF path of hemostasis, but the damage extended to ECs and SET/EVT produces fibrin clots via fibrinogenesis and macrothrombosis via macrothrombogenesis by combined activation of ULVWF path and TF path as shown in Figure 1. Microthrombosis (i.e., VMTD) is the disease of the microvasculature (e.g., capillaries, arterioles, venules and sinusoids) due to activated ULVWF path. However, macrothrombosis is always the disease of the macrovasculature due to activated ULVWF path and TF path in local or regional vascular trauma.

The extent of vascular wall damage limited to ECs leading to endotheliopathy as seen in sepsis is commonly disseminated, but occasionally may cause, focal, local, multifocal, regional EA-VMTD. In contrast, vascular wall damage in local vascular trauma is localized or regionalized and causes macrothrombosis (e.g., DVT). Macrothrombus is always local or regional disease but has propensity to travel to the lungs as emboli to produce PTE. Succinctly, vascular wall damage creates the character of microthrombosis, fibrin clots disease and macrothrombosis, and vascular tree damage determines disseminated or localized/regionalized character of thrombosis.

Because of the efferent travel of arterial blood from the heart, arterial microthrombosis causes arteriolar and capillary microthrombosis (e.g., diffuse encephalopathic stroke, MODS), but in the afferent travel of venous blood to the heart, venous microthrombosis leads to pulmonary capillary microthrombosis in the lungs (e.g., ARDS). Arterial microthrombosis leads to organ hypoxia (i.e., MODS) and TTP-like syndrome, but venous microthrombosis leads to ARDS and ITP-like syndrome [10,11] as summarized in Figure 4 and Table 4. The microthrombi strings are composed of binary components of platelet-ULVWF/FVIII complexes, and fibrin clots are composed of fibrin meshes/NETs (Figure 1). Thus, macrothrombus unified of microthrombi and fibrin meshes/NETs are composed of ternary components of platelets, ULVWF/FVIII and fibrin meshes/NETs [25].

### 4.3. Factors Caused by Associated Underlying Disease or Another Vascular Event

The most common thrombosis encountering among many human diseases is microthrombosis (i.e., VMTD), which can be induced by sepsis (e.g., bacterial, viral, fungal, rickettsial and parasitic), polytrauma, envenomation, side effects of drugs, diabetes, autoimmune disease, and complications of surgery, pregnancy and transplant. EA-VMTD is usually disseminated in the microvasculature [14,21,22,23,24,25,26,27]. It is well established to be due to complement activated endotheliopathy leading to ECs dysfunction [28,29]. In the past and still now, the diagnosis of EA-VMTD has been masked or wrongly termed as disseminated coagulopathy (e.g., DIC) [21,22]. In addition to the concept of EA-VMTD, aEA-VMTD and vEA-VMTD arising from two vascular milieux are now recognized as two different phenotypic entities of microthrombosis after the experience learned from COVID-19 pandemic [10,11,25].

Intravascular trauma or injury (e.g., accident, vascular access, vascular device, and surgery) leading to macrothrombosis is common, especially in the hospital and ICU, which includes DVT, VTE, and catheter-associated thrombosis. The localized thrombosis may be the result of external or internal bodily injury and intravascular injury. Should a critically ill patient with underlying microthrombosis (e.g., sepsis, diabetes or cancer) be admitted to the hospital and undergo diagnostic and therapeutic access utilizing the vascular device or vessel injuring procedure (e.g., surgery), intravascular fibrin meshes can be formed due to activated TF path leading to fibrinogenesis. In this case, the interaction of microthrombi strings and fibrin meshes can come together and produces a more complex form of blood clot(s) disease via the unifying mechanism, which creates a complex phenotype, called combined micro-macrothrombosis (e.g., VTE, SPG) as illustrated in Figure 4 [11]. Every different thrombotic disorder except fibrin clot disease has been well-documented in COVID-19 pandemic, which examples and their pathogeneses are shown in Figure 5. Four different thrombogenetic mechanisms: microthrombogenesis, fibrinogenesis, macrothrombogenesis, and combined micro-macrothrombogenesis, succinctly explain every and each thrombotic syndrome occurring in COVID-19 sepsis [11,14,25]. An underlying disease (e.g., EA-VMTD) and another vascular injury (e.g., vascular injury in ICU) can interact in different vascular milieux and produce several complex forms of the clinical phenotype such as VTE, PTE, IVCT/SVCT, SVT, purpura fulminans, diabetic gangrene, SPG, Fournier’s disease, acute necrotizing fasciitis. Previously and still now, some clinicians have called arterial combined micro-macrothrombosis causing peripheral gangrene syndrome as “DIC” [25].

Endotheliopathy is a very colorful pathologic disorder with well-established molecular pathogenesis (Figure 2), creating the complex phenotypes of the microthrombotic disorder as well as inflammation and autoimmune phenomenon [25]. Finally, with novel classification of thrombotic disorders, the diagnosis of endotheliopathy can be readily recognized by endothelial molecular markers and laboratory tests, which includes activated complement components, increased cytokines, markedly increased FVIII activity and overexpressed ULVWF/VWF antigen, and consumptive thrombocytopenia [10,11]. Among these markers, inflammatory markers (i.e., cytokines) activate inflammatory pathway provoking inflammation and multiorgan inflammatory syndrome (MOIS) [13,14,21], and microthrombotic markers (i.e., overexpressed ULVWF/FVIII and platelets) trigger microthrombotic pathway promoting microthrombogenesis [13,14,25,26,27] orchestrationg EA-VMTD and MODS, and both pathways also contribute to autoimmune pathway [25,28,29,30,31] promoting to autoimmunity and autoimmune diseases.

### 4.4. Factors Caused by Tropism, Endothelial Heterogeneity, Environmental and Gene Interaction

In addition to above mentioned factors, endotheliopathy can be taylored by endothelial heterogeneity in different person by the same pathogen and may modify thrombogenesis, changing the clinical phenotype expression of thrombosis [14,15] as the tropism of pathogen influences human tissue/organ susceptibility [32,33,34]. The endothelial heterogeneity of host [21,35,36,37] would play a role in microthrombosis more than in macrothrombosis. Also, tropism is known to play a significant role producing in clinical phenotypes of microthrombotic disorder such as Waterhouse-Friderichsen syndrome by meningococcus’s propensity attracting to the adrenal glands [38], ARDS by SARS-CoV-2′s predilection for the lungs [32,33], cerebral malaria by plasmodium falciparum’s attraction to the brain [39]. Perhaps, the endothelial heterogeneity of host may play a serious role producing posttransplant veno-occlusive disease to the liver [37,40], and Ebola viral hemorrhagic disease leading to hepatic coagulopathy [24]. In addition to environmental factors such as aging, life style, high altitude travel and smoking, macrothrombosis may be upregulated by genetic factors, especially thrombophilia affecting TF pathway (e.g., gene mutation of protein C and protein S, and factor V Leiden) [41], and microthrombosis can be upregulated by thrombophilia affecting ULVWF pathway (e.g., ADAMTS13 polymorphism) [23].

### 4.5. Thrombosis Due to Aberrant Hemostasis without Vascular Injury

Unlike thrombosis caused by intravascular injury, atypical thrombotic syndromes without vascular injury [4] have been associated with activated aberrant ULVWF path in TTP due to severe ADAMTS13 deficiency leading to microthrombosis (i.e., hereditary TTP and acquired antibody-associated TTP) [23], activated aberrant TF path in fibrin clot disease of APL due to pathologic expression of TF in leukemic cells [42]. Another non-hemostatic thrombosis is “platelet thrombosis” in HIT-WCS [43]. HIT-WCS is not caused by thrombogenesis via hemostatic mechanism because there is no evidence it utilizes either ULVWF path and/or TF path.

Following heparin administration in a critically ill patient admitted to hospital or ICU, heparin-platelet factor 4 (PF4) complex may be formed, which can become a specific antigen. This heparin-PF4 antigen produces heparin-PF4 antibody [44] in a patient. When heparin is continued or reintroduced, heparin-PF4 antigen and antibody complexes are formed, which activate platelets, leading to intravascular aggregation of platelets (white clot), and produce the multiple large and long platelet thrombi [45]. This syndrome can be called truly “platelet thrombosis” occurring in both venous and arterial systems, especially in lower extremities. This has been termed “white clot syndrome”. Recently observed HIT-like syndrome with thrombosis in COVID-19 infection and vaccination in the absence of heparin exposure is not true HIT-WCS because ant-PF4 antibody is non-specific antibody and the thrombus is not true “white clots”. In this case, anti-PF4 antibody is an epiphenomenon unrelated to heparin. In SARS-CoV-2 infection and vaccination, VTE and thrombocytopenia are not caused PF4 antibody [25]. Thrombocytopenia is consumptive one due to endotheliopathy, and VTE is the result of venous combined micro-macrothrombosis due to underlying endotheliopathy and additional vascular injury from ICU as explained earlier and illustrated in Figure 4 and Figure 5 [10,11].

## 5. ADAMTS13 Role in Thrombogenesis

ADAMTS13 is probably the most important enzyme modulating hemostasis by preventing low grade thrombogenesis and inhibiting progressive atherosclerosis. The enzyme maintains normal cardiovascular physiology in human through the control of ULVWF path of hemostasis lifetime. This enzyme may protect the vascular integrity and health globally for human, but its insufficiency affects negatively for the long term [46,47] and significantly contributes to thrombogenesis for the short term in critically ill patients [48]. ADAMTS13 is primarily synthesized in the stellate cells of the liver [49], and its function is to cleave ULVWF multimers anchored on the endothelial surface to prevent microthrombogenesis at the site of vascular injury [21]. The deficiency of plasma ADAMTS13 activity less than 10 percent of normal due to mutations of the gene or antibodies against the enzyme occurs in hereditary TTP (gene mutation-associated: GA-VMTD) and acquired VMTD (antibody-associated: AA-VMTD).

ADAMTS13 activity may be mild to moderately reduced between 20 to 75% in some individuals due to heterozygous gene mutation or polymorphism. Their life is typically undisturbed in the absence of critical illnesses. However, when this individual with borderline low level of ADAMTS13 develops sepsis due to pathogen, becomes pregnant with complication or is injured by polytrauma, ADAMTS13 insufficiency may not be able to handle the excess of ULVWF released from ECs, and can trigger the activation of ULVWF path of hemostasis and produce microthrombosis. Thus, EA-VMTD may cause the same triad of consumptive thrombocytopenia, MAHA, and MODS seen in TTP [23]. Therefore, this condition is called TTP-like syndrome. Although the primary role of ADAMTS13 is cleaving ULVWF multimers to smaller molecular weight VWF to prevent microthrombogenesis, its insufficiency can have a far-reaching effect by upregulating macrothrombogenesis via “two-path unifying theory” of hemostasis and by promoting combined micro-macrothrombogenesis (Figure 4). Indeed, ADAMTS13 insufficiency has been well documented in DVT and VTE [50,51,52,53,54].

In intravascular injury, the major biologic molecule controlling the rise and fall of thrombogenenesis is the protease ADAMTS13 which downregulates microthombogenesis. This deficiency promotes microthrombi strings composed of platelet-ULVWF/FVIII complexes that unify with fibrin meshes and contribute to VTE [10,11]. Thus, the sufficient level of ADAMTS13 restrains progressive atherosclerosis and maintains good vascular health of an individual for the long term and also prevents unnecessary thrombogenesis of both microthrombosis and combined micro-macrothrombosis.

## 6. Current Classification of Thrombotic Disorders without Molecular Mechanism

The term thrombogenesis can be defined as a process of forming variable phenotypes of thrombosis initiated by hemostasis in intravascular injury, which leads to two subhemostatic paths producing thrombotic disorders via interacting mechanisms with underlying disease, environmental and genetic factors. Every phenotype of thrombosis except previously mentioned non-hemostatic disorders is the result of hemostasis in intravascular injury that leads to the same molecular thrombogenesis. Because of our poor comprehension of the mechanism of hemostasis and thrombogenesis, we have used the clinical terms on thrombotic disorders without rational classification as exampled below.

### 6.1. Currently Often Used Terms for the Phenotype

Commonly designated thrombosis terms include:Vessel milieu-designated thrombosis type
Deep venous thrombosis (e.g., distal DVT, VTE, PTE)Superficial vein thrombosisArterial thrombosisMicrovascular thrombosis (e.g., TTP-like syndrome)Venous sinusoidal thrombosis (e.g., VOD)“DIC”Dysfunction expressive type
Stroke (e.g., transient ischemic attack [TIA], AIS)Heart attack (e.g., Acute myocardial infarction)CoagulopathyPelvic congestion syndromeGangreneThrombo-hemorrhagic syndromeVascular localization type
Cerebral artery thrombosisAortic arch thrombosisCerebral venous sinus thrombosisCoronary artery thrombosisRenal vein thrombosisPTEVTELocalized venous thrombosis (e.g., IVCT/SVCT, PVT, BCS, SVT)Symmetrical peripheral gangrene (SPG)Limb gangreneOrgan localization type (but often the thrombotic nature is not recognized.)
EncephalopathyPTEAcute respiratory distress syndrome (ARDS)Acute necrotizing pancreatitisAcute liver failureHemolytic-uremic syndromeRhabdomyolysisMultiorgan dysfunction syndrome (MODS)Pathologic expressive type
TTP-like syndrome (e.g., arterial endotheliopathy)ITP-like syndrome (e.g., venous endotheliopathy)Acute necrotizing fasciitisDiabetic gangreneUndefined clinical syndrome type
Heyde’s syndromeSusac syndromeHemolytic-uremic syndromeHereditary hemorrhagic telangiectasiaPurpura fulminans

These designations typically represent simplified and expressive phenotypes of thrombotic disorders based on clinical and pathologic manifestations without the consideration of involved hemostasis, thrombogenesis and pathogenesis. Because of this reason, thrombosis often has not been considered as a major issue over the primary underlying diseases or conditions (e.g., sepsis, polytrauma, autoimmune disease, cancer, complications of pregnancy, transplant and surgery), but secondary incidental disorder. To the contrary, it should be emphasized that every thrombotic disease has major impact to the well-being of the patient through its pathogenesis and significantly undermines the outcome of the underlying disease. Thus, for the proper diagnostic, prognostic and therapeutic approaches, it is essential to classify thrombotic disorders based on molecular hemostasis.

### 6.2. Proposed Novel Classification of Thrombotic Disorders Based on Thrombogenesis and Intrinsic Character

To distinguish different phenotypes of thrombosis based on hemostasis and mechanisms of thrombogenesis and to identify the underlying disease and search for effective therapeutic regimens, novel classification system is proposed as presented in Table 5. Four basic characters of primary phenotypes derived from three thrombogenetic mechanisms are: (1) microthrombosis composed of microthrombi strings of platelet-ULVWF/FVIII complexes, (2) fibrin clot disease made of fibrin meshes and thrombin, (3) macrothrombosis formed of macrothrombus containing microthrombi-fibrin meshes, and (4) combined micro-macrothrombosis composed of microthrombi-fibrin meshes from two different vascular injuries. However, the clinical phenotypes can become more complicated if two different vascular milieux, arterial and venous system are included. The followings are primary phenotypes of thrombosis based on their intrinsic characters.

#### 6.2.1. Microthrombosis

All of the microthrombosis is called VMTD, which includes TTP (i.e., gene mutation-associated-VMTD, antibody-associated-VMTD), TTP-like syndrome (EA-VMTD) and focal, local, multifocal, and regional EA-VMTD [23]. VMTD is a newly established disease entity of pathologic and clinical phenotype among thrombotic disorders following discovery of “two-activation of the endothelium” and “two-path unifying theory” of hemostasis [3,4]. The term microthrombosis has been used in the medical literature when blood clots were present in arterioles and capillaries as hyaline (micro)thrombi since the time of Donald McKay and recently [55,56]. In the past, the term microthrombi have been used interchangeably with that of fibrin clots not only in TTP but also in “disseminated intravascular coagulation” (“DIC”) [56] until this author published the character of “microthrombi strings” composed of the “platelet-ULVWF complexes” [3,4]. The partial hemostatic mechanism of ULVWF path producing microthrombosis via microthrombogenesis has been illustrated in Figure 1 and Figure 2. Since partial hemostasis occurs in sepsis due to disseminated endotheliopathy (i.e., ECs injury) without SET/EVT injury, microthrombosis is concluded to be the result of lone activation of ULVWF path. Because of its disseminated feature, EA-VMTD has been a deadly disease, which sometimes has been called “DIC”, thrombotic microangiopathy or acquired “TTP” [22,57]. Now the different concepts of microthrombosis as VMTD in TTP and endotheliopathy-associated microthrombosis as EA-VMTD representing TTP-like syndrome are well established [23]. The pathogenesis of EA-VMTD is shown in Figure 2.

Microthrombosis (i.e., VMTD) is clearly different disease from fibrin clot disease and macrothrombosis. As summarized in Table 5, although the character of microthrombi strings is the same, the clinical phenotypes are remarkably different in arterial and venous milieux, large vasculature and microvasculature, and amongst focal, local, multifocal, regional and generalized involvement, which examples are listed in Figure 4 and Table 4.

#### 6.2.2. Macrothrombosis

Macrothrombosis is a thrombotic disease formed of the binary complex of “microthrombi strings” from microthrombogenesis and “fibrin meshes” from fibrinogenesis due to vascular wall damage extending from ECs to SET/EVT via the unifying mechanism following an intravascular injury, which is called “macrothrombogenesis [2,3,21]. This disease includes arterial macrothrombosis and isolated venous macrothrombosis representing aortic thrombosis, AIS, AMI, and distal DVT (Table 5). This pathogenesis is simple and straightforward and almost always caused by a local vascular injury. Macrothrombosis is localized without activation of inflammatory pathway.

#### 6.2.3. Fibrin Clot Disease

Unlike VMTD or macrothrombosis, fibrin clot disease should be defined as true clotting disorder (coagulopathy) made of fibrin clots within intravascular space as seen only in APL without an intravascular injury. The patient with APL [58,59,60] was found to overexpress TF in leukemic promyelocytic cells. When TF activates FVII and form TF-FVIIa complex, TF path (Figure 1) promotes sequential activation of coagulation proteins and serine proteases via the extrinsic clotting cascade, which is not detailed in the Figure because this coagulation scheme is well known. In TF pathway, TF-FVIIa activates FIX to FIXa to form FVIII-FIXa complex (tenase), which activates to FX to FXa [61]. FXa activates FV to FVa, in which FXa and FVa form FXa-FVa complex (prothrombinase) [62] that converts prothrombin to thrombin and facilitates fibrinogen to fibrin clot formation. Since the vascular endothelium (i.e., ECs) is not compromised in APL, ULVWF path of hemostasis is not activated.

Therefore, this disorder is not called as thrombotic disorder because platelets are not participated in fibrinogenesis and fibrin clots are not true thrombus, and is neither microthrombosis nor macrothrombosis. Thus, it should be called disseminated “fibrin clot disease” with consumption of FVIII and FV and secondary fibrinolysis. The clinical manifestation of “fibrin clot disease” is also “hemorrhagic disease” due to depleted/inactivated coagulation factors (i.e., FI, FV and circulating FVIII) as shown in Table 6 [63]. Thrombocytopenia in APL is not due to consumption of platelets occurring in endotheliopathy but is the result of bone marrow suppression from leukemia.

The diagnosis for coagulopathy can be established with the findings of hypofibrinogenemia, decreased activity of FVIII and FV due to consumption (Table 6) in a hemorrhagic patient with the diagnosis of APL. Clinically it is not thrombosis, but can be called true DIC—if we insist the term DIC is a clinical diagnosis. Although it is the result of aberrant partial hemostasis (fibrinogenesis) without intravascular injury, still, fibrin clot disease may be included in “thrombosis” column for educational purpose as an oddity (Table 5).

#### 6.2.4. Combined Micro-Macrothrombosis

I have made an interpretation on this complex thrombosis syndrome in several articles before and during COVID-19 pandemic [10,11,14,25]. Combined micro-macrothrombosis represents many complex phenotypes that can occur in both arterial and venous milieux with striking clinical features, typically occurring in hospital, especially in the ICU setting.

Clinical phenotypes of this thrombosis develop in the patient with underlying diagnosis of microthrombosis due to endotheliopathy (i.e., EA-VMTD) seen in sepsis or other critical illnesses [25] and additional fibrin meshes resulting from another vascular injury due to vascular accesses in hospital or ICU. In clinical practice, venous combined micro-macrothrombosis (e.g., VTE) is much more common than arterial combined micro-macrothrombosis (e.g., SPG) because the central venous catheter and venous access are more commonly utilized on the venous system in hospital and ICU. These disorders are produced by the unifying mechanism of “microthrombi strings” from EA-VMTD and “fibrin meshes” from another vascular injury in hospital/ICU according to “two-path unifying theory” of hemostasis. These phenotypes are the results of “combined micro-macrothrombogenesis”.

The complex micro-macrothrombosis can occur in the arterial system and/or in the venous system with two distinctly different clinical phenotypes [10,11]. Arterial microthrombi strings could encounter fibrin meshes produced by additional traumatic arterial injury (e.g., indwelling arterial catheter) while transitioning efferently in blood circulation away from the heart. In this situation, the microthrombi strings and fibrin meshes would produce numerous minute “combined micro-macrothrombi” of ternary structure (i.e., platelets, ULVWF/FVIII and fibrin meshes) by the unifying mechanism similar to macrothrombogenesis shown in Figure 1. It is likely that combined ternary micro-macrothrombi are minute and similar-sized. When these minute macrothrombi made of micro-macrothrombi are occurred at the peripheral small arterial vasculatures, especially in digits, they become a shower of minute thrombi and completely occlude smaller peripheral arteries at similar sized vasculatures. This causes peripheral total anoxia of digits without collateral circulation, which would lead to total denaturation of trapped hemoglobin molecules to methemoglobin and organic ferric sulfide turning to dark, dry and mummified gangrene of the peripheral tissues. The typical cases are SPG [64,65,66,67] and limb gangrene [68,69].

On the other hand, in “silent” microthrombi (ITP-like syndrome) in the venous system, “microthrombi strings” would encounter “fibrin meshes” formed at additional venous injury site (e.g., indwelling central venous catheter) and form multiple, irregular, large, and connected venous “combined ternary micro-macrothrombi” along with NETs at the venous injury site due to slow venous circulation. These localized or regionalized combined venous micro-macrothrombi (i.e., VTE) travel afferently to the heart and lungs causing pulmonary thromboembolism (PTE) [10,11]. In combined arterial and/or venous micro-macrothrombosis, inflammation can be a significant component due to associated underlying endotheliopathy [70]. It is no wonder why both ARDS (i.e., microthrombosis) and VTE/PTE (i.e., combined micro-macrothrombosis) commonly coexist in a COVID-19 patient admitted to ICU.

#### 6.2.5. Interpretation for Thrombotic Disorders as an Example in COVID-19 Infection

In COVID-19 pandemic, the major mystery has been why some patients have developed several different thrombotic disorders, including ARDS, various forms of DVT such as VTE, PTE, SVT, IVCT, SVCT, BCS, PVT, arterial thrombosis, venous thrombosis, stroke, myocardial infarction, arterial gangrene syndromes such as SPG, limb gangrene, acrocyanosis, purpura fulminans, Fournier’s gangrene, peripheral digit ischemia, COVID toes [71,72,73,74,75,76,77,78]. Additionally, rare cases of TTP-like syndrome [79,80,81,82] associated with aEA-VMTD and ITP-like syndrome [10,11] suspected to be associated with vEA-VMTD have occurred in COVID-19 infection likely due to different vascular milieux. Prior to COVID-19 pandemic, I published an unequivocal evidence of microthrombotic nature of ARDS occurring in sepsis caused by pathogen, including SARS-CoV and MERS-CoV [13]. In the past, without the molecular mechanism of hemostasis in vivo, it was an impossible task to identify the concept and pathogenesis of microthrombogenesis amongst the different paths of thrombogenesis.

Once “two-path unifying theory” of hemostasis and “two-activation theory of the endothelium” are understood, the interpretation of every thrombotic disorder encountered in COVID-19 infection could be easily inferred and logically explained. The followings are essential components in identifying the thrombogenetic mechanisms of COVID-19 infection as illustrated in Figure 5.

COVID-19 sepsis produces endotheliopathy via complement activation perhaps induced by S protein and/or other toxins of SARS-CoV-2 [28,83].Endotheliopathy leading to ECs damage activates inflammatory pathway and microthrombotic pathway (ULVWF path) as shown in Figure 2 [13,21].Endotheliopathy causes inflammation via cytokines release and VMTD via ULVWF/FVIII release [13,21].ULVWF/FVIII becomes anchored to endothelial membrane of the ECs [84,85,86,87].ULVWF strings would recruit platelets and assemble “microthrombi strings” composed of the platelet-ULVWF complexes at terminal arterioles, capillaries, venules and sinuses [21,88].In COVID-19 sepsis, disseminated EA-VMTD (microthrombosis) would occur rarely in the arterial system, but often in the venous system. TTP-like syndrome is the result of aEA-VMTD and ITP-like syndrome is the result of vEA-VMTD [10,11].Often vascular access, devices and mechanical ventilation would be needed in ICU for critically ill COVID-19 patients, which can contribute traumatic intravascular injury, culminating to combined micro-macrothrombosis [14].Traumatic injury penetrating to SET/EVT is suspected to release TF and activate TF path continuously in circulation and forms “fibrin meshes” via TF-FVIIa complex activating coagulation cascade, especially from the vascular device insertion area [11].Unifying mechanism of “microthrombi strings” and “fibrin meshes” produces the ternary complex of “micro-macrothrombi” leading to combined arterial micro-macrohrombosis (e.g., SPG) and/or venous micro-macrothrombosis (e.g., VTE) [10,11,14,25].

The thrombogenetic mechanism of every clinical and pathologic phenotypes occurring COVID-19 sepsis is depicted in Figure 5. It is self-explanatory.

## 7. Diagnosis and Therapeutic Perspectives

Each clinical and pathologic phenotype for every thrombosis could be determined by following factors as applied in Table 3 and Figure 5.

Different vascular milieux (arterial vs. venous)Depth, extent and site of vascular damage (ECs vs. ECs,/SET/EVT)Activated hemostasis and subsequent thrombogenesis in different-sized vasculatures (microvasculature vs. macrovasculature)Physiologic function of involved organ and tissue (e.g., brain vs. muscle)Interaction with environmental, genetic, and underlying pathologic factors

To aid in the differential diagnosis and to help for the comprehension of thrombogenetic disparity amongst several different primary phenotypes of thrombosis, summarized in Table 6 is the expected laboratory data based on hemostatic theories. As explained in the text, The most of data are congruous between actual, clinical and laboratory data on each thrombosis reported in the medical literature and theory-based and anticipated clinical and pathologic data in different phenotypes of thrombosis. These findings also validate novel two hemostatic theories and the “unifying mechanism”.

Therapeutic approach based on novel classification of the thrombotic disorder could be discussed in an appropriate forum(s) amongst coagulation specialists, including the following issues.

In critically ill patients in hospital or ICU, early diagnosis of EA-VMTD is the first step preventing the complex forms of thrombotic disorders and then implementing proper therapy with antimicrothrombotic regimens.The effective prevention of macrothrombosis is to avoid vascular injury in the hospital and ICU with minimum vascular access and care. Its thromboprophylactic measure may provide only a limited value as long as the underlying endotheliopathy is persistent.Therapeutic regimens for EA-VMTD may include plasma exchange and clinical trials with antimicrothrombotic agents (e.g., rADAMTS13, N-acetyl cysteine) and complement inhibitors.The most effective approach preventing combined micro-macrothrombosis is diligence with absolute minimum of vascular access and intervention, and recognizing the potential of the deadly complications of VTE and gangrene syndromes.

## 8. Conclusions

All thrombotic disorders occur as a result of normal hemostasis after an intravascular injury except few conditions (i.e., TTP, APL and HIT-WCS). To date, the pathogenesis of thrombotic disorders has not been clearly established because in vivo hemostasis has been incompletely identified. However, new concepts of in vivo hemostasis and endotheliopathy based on the vascular wall model causing vascular damage have identified the true nature of thrombogenesis. This includes microthrombogenesis initiating “microthrombi strings”, fibrinogenesis promoting “fibrin meshes”, and the unifying mechanism of these molecules producing macrothrombosis via macrothrombogenesis. Based on this concept of thrombogenesis, three primary phenotypes of thrombosis are discovered and seven secondary phenotypes are identified. Further, additional phenotypes that are influenced by interaction with the underlying diseases, and environmental and genetic factors are being recognized. Armed with two hemostatic theories, hemostatic fundamentals and knowledge on the molecular mechanisms of thrombogenesis, novel phenotype classification should open a new frontier in thrombosis research and studies on the entire spectrum of cardiovascular disease. It is a high time to return to the basics of hemostasis not only in redesigning therapeutic regimens of various thrombotic disorders, but also implementing proper measures to prevent life-threatening thrombotic disorders.

## Figures and Tables

**Figure 1 biomedicines-10-02706-f001:**
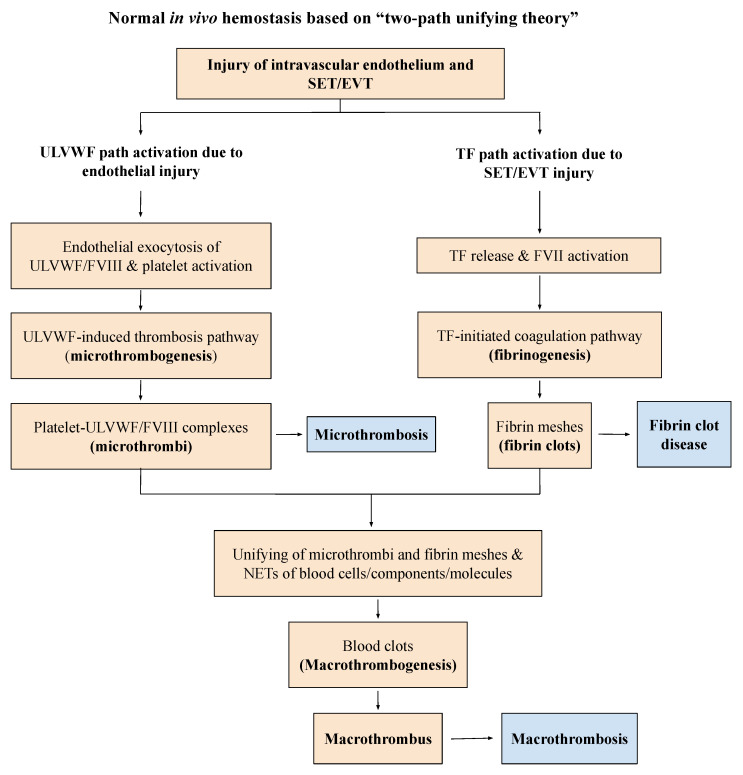
Normal hemostasis in vivo based on “two-path unifying theory” (Reproduced and modified with permission from Chang JC. *Thrombosis Journal*. 2019;17:10). Following a vascular injury, hemostatic system in vivo activates two independent sub-hemostatic paths: microthrombotic (ULVWF path) and fibrinogenetic (TF path). The former is initiated by the damage of ECs and the latter by that of SET/EVT in external bodily injury and intravascular injury. In activated ULVWF path, ULVWF/FVIII are released and recruit platelets, and produce microthrombi strings via microthrombogenesis, but in activated TF path, TF is released and activates FVII. The TF-FVIIa complex produces fibrin meshes via fibrinogenesis of extrinsic coagulation cascade. The final path of in vivo hemostasis is macrothrombogenesis, in which microthrombi strings and fibrin meshes become unified together with the incorporation of NETs, including red blood cells, neutrophils, DNAs and histones. This unifying event “macrothrombogenesis” promotes the hemostatic plug and wound healing in external bodily injury and produces macrothrombus causing macrothrombosis in intravascular injury. Abbreviations: EA-VMTD, endotheliopathy-associated vascular microthrombotic disease: ECs, endothelial cells; EVT, extravascular tissue; NETs, neutrophil extracellular traps; SET, subendothelial tissue; TF, tissue factor; ULVWF, ultra large von Willebrand factor multimers.

**Figure 2 biomedicines-10-02706-f002:**
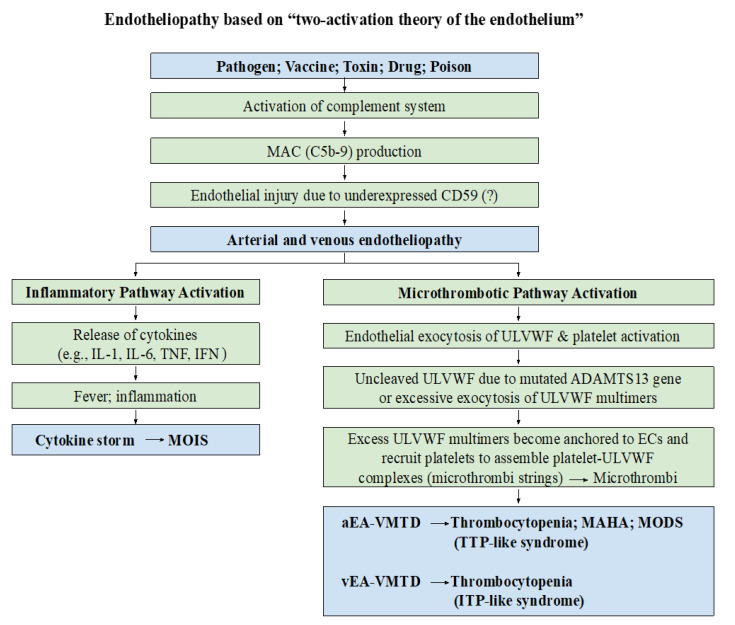
Pathogenesis of endotheliopathy based on “two-activation theory of the endothelium”. Endothelial molecular pathogenesis is initiated by the activated complement system following exposure to pathogen, toxin, drug, poison, venom, vaccine, polytrauma, hyperglycemia, severe hypertension, and others. Endotheliopathy releases inflammatory cytokines and hemostatic factors, and activates two clinically distinctive molecular pathways: inflammatory and microthrombotic. Both arterial endotheliolpathy and venous endotheliopathy provoke inflammation via inflammatory pathway leading to the inflammatory syndrome called MOIS due to cytokines, but arterial endotheliopathy promotes microthrombosis via microthrombotic pathway and produces aEA-VMTD due to activation of ULVWF path of hemostasis (shown in Figure 1). Arterial endotheliopathy is characterized by the triad of consumptive thrombocytopenia, MAHA, and MODS, which is called “TTP-like syndrome”, but venous endotheliopathy is characterized by ITP/“ITP-like syndrome” due to silent microthrombi with consumptive thrombocytopenia as explained in the text. The distinguishing features are caused by different anatomy, physiological function and hemodynamic characteristics between arterial system and venous system. These are very important pathological features in the understanding of the complexity of variable thrombotic phenotypes occurring endotheliopathic syndromes. Abbreviations: EA-VMTD, endotheliopathy-associated vascular microthrombotic disease; ITP, immune thrombocytopenic purpura; MAC, membrane attack complex; IFN, interferon; IL, interleukin; MAHA, microangiopathic hemolytic anemia; MODS, multiorgan dysfunction syndrome; MOIS, multiorgan inflammatory syndrome; TNF, tumor necrosis factor; TTP, thrombotic thrombocytopenic purpura; aEA-VMTD, arterial EA-VMTD; vEA-VMTD, venous EA-VMTD; ULVWF, ultra large von Willebrand factor.

**Figure 3 biomedicines-10-02706-f003:**
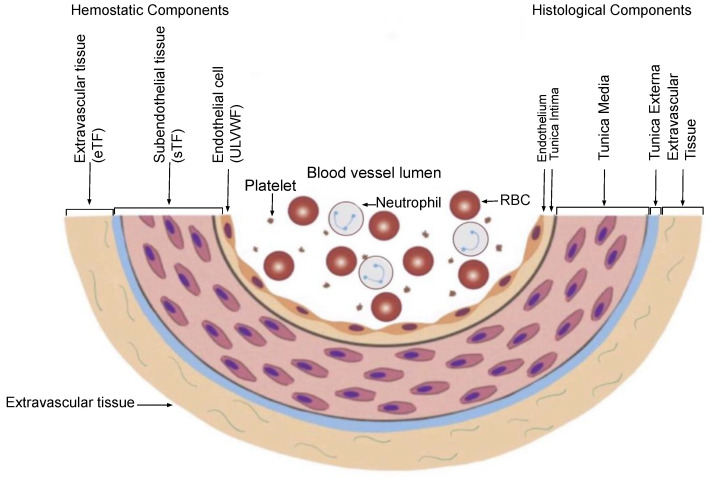
Schematic illustration of cross section of blood vessel histology and hemostatic components (Reproduced and modified with permission from Chang JC. *Clin Appl Thromb Hemost* 2019 Jan-Dec; 25:1076029619887437). The blood vessel wall is the site of hemostasis (coagulation) in vivo to produce blood clots (hemostatic plug) and stop hemorrhage in the external bodily injury. It is also the site of hemostasis (thrombogenesis) to produce intravascular blood clots (thrombus) in the intravascular injury to cause thrombosis. Its histologic components are divided into the endothelium, tunica intima, tunica media and tunica externa, and each component has its function contributing to molecular hemostasis. As illustrated, ECs damage triggers the exocytosis of ULVWF/FVIII and SET damage promotes the release of sTF from the tunica intima, tunica media and tunica externa. EVT damage releases of eTF from the outside of blood vessel wall. This depth of blood vessel wall injury contributes to the genesis of different thrombotic disorders such as microthrombosis, macrothrombosis, fibrin clots, thrombo-hemorrhagic clots and various endotheliopathic syndromes. This concept based on the blood vessel wall model is especially important in the understanding of different phenotypes of stroke and heart attack. Abbreviations: EVT, extravascular tissue; eTF, extravascular tissue factor; SET, subendothelial tissue; sTF, subendothelial tissue factor; RBC, red blood cells; ULVWF, ultra large von Willebrand factor.

**Figure 4 biomedicines-10-02706-f004:**
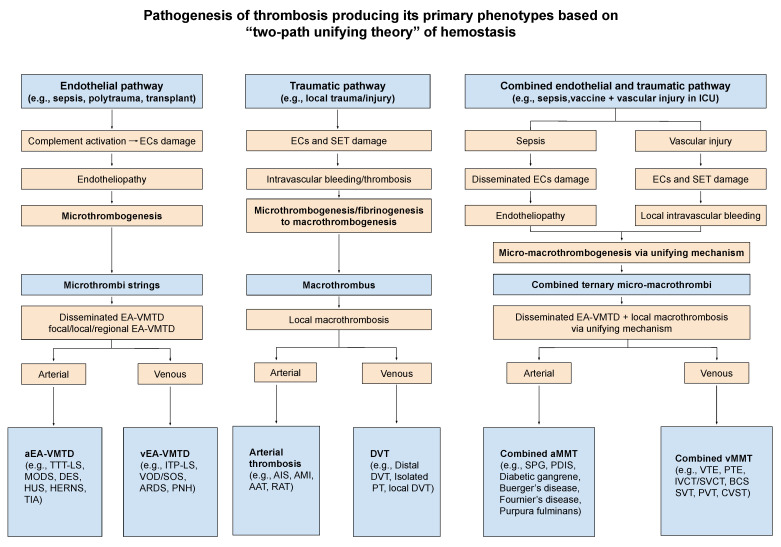
Pathogenesis of thrombosis producing its primary and secondary phenotypes based on “two-path unifying theory of hemostasis. Thrombogenesis following an intravascular injury may take two different paths. For example, in sepsis the intravascular injury is limited to ECs but disseminated. The damage releases ULVWF/FVIII and activates ULVWF path (microthrombotic pathway) that produces microthrombi strings, eventually leading to microthrombosis (e.g., aEA-VMTD). However, in a local trauma the intravascular injury is extended from ECs to SET/EVT but localized, which releases TF and small amount of ULVWF/FVIII and activates TF path (traumatic pathway) that produces blood clots composed of fibrin meshes, small amount of ULVWF/FVIII and platelets, and NETs, leading to macrothrombosis (e.g., DVT). Sometimes the septic patient with disseminated microthrombi strings due to endotheliopathy admitted to ICU could also develop another unrelated serious vascular damage to SET/EVT due to an indwelling vascular device in ICU care, which releases TF and promote fibrinogenesis that forms fibrin meshes. In this case, combined micro-macrothrombi composed of ternary components of ULVWF/FVIII, platelets, and fibrin meshes could be formed and lead to combined micro-macrothrombosis shown in the Figure with clinical phenotypes. The thrombosis due to EA-VMTD and another vascular injury is a complex form of micro-macrothrombosis. Their thrombogeneses are logical and has been explained in the text and my previous publications [14,25]. Abbreviations: AAT, aortic arch thrombosis; aEA-VMTD, arterial endotheliopathy-associated vascular microthrombotic disease; AIS, acute ischemic stroke; AMI, acute myocardial infarction; ARDS, acute respiratory distress syndrome; BCS, Budd-Chiari syndrome; DES, diffuse encephalopathic stroke; CVST, cerebral venous sinus thrombosis; DVT, deep venous thrombosis; ECs, Endothelial cells; EVT, extravascular tissue; HERNS, hereditary endotheliopathy, retinopathy, retinopathy and stroke syndromes; HUS, hemolytic-uremic syndrome; ICU, intensive care unit; ITP-LS, ITP-like syndrome; IVCT/SVCT, inferior vena cava thrombosis/superior vena cava thrombosis; MODS, multiorgan dysfunction syndrome; NETs, neutrophil extracellular traps; PDIS, peripheral digit ischemic syndrome; PNH, paroxysmal nocturnal hemoglobinuria; PT, pulmonary thrombosis; PTE, pulmonary thromboembolism; PVT, portal vein thrombosis; RAT, renal artery thrombosis; SET, subendothelial tissue; SOS, sinus obstruction syndrome; SPG, symmetrical peripheral gangrene; SVT, splanchnic vein thrombosis; TF, tissue factor; TIA, transient ischemic attack; TTP-LS, TTP-like syndrome; ULVWF, ultra large von Willebrand factor; VOD, veno-occlusive disease; VTE, venous thromboembolism.

**Figure 5 biomedicines-10-02706-f005:**
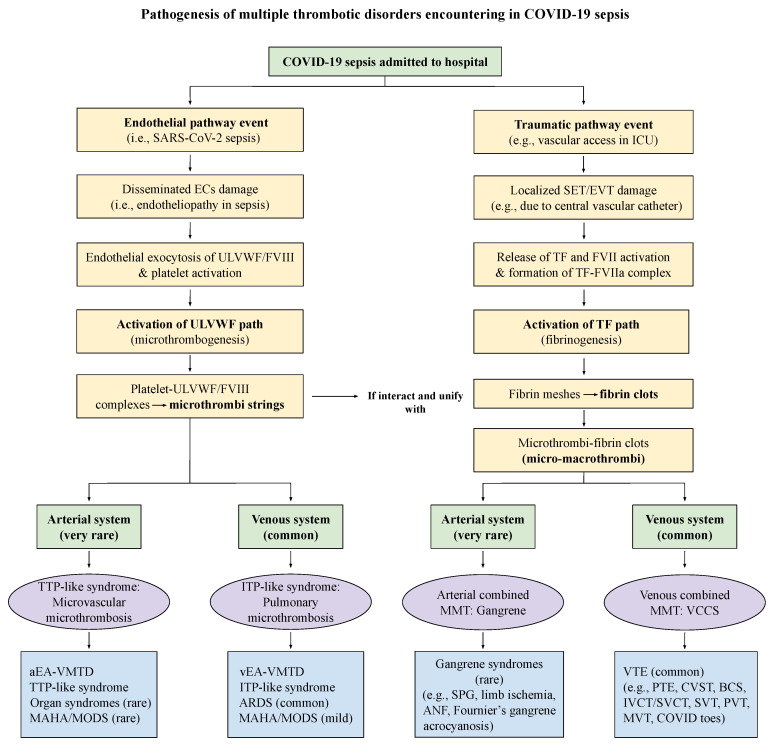
Pathogenesis of multiple thrombotic disorders encountering in COVID-19 sepsis. Unlike other sepsis, including bacterial, viral, fungal, rickettsial and parasitic sepsis, COVID-19 has been highlighted by the mystery of extensive thrombotic diseases, which has included ARDS, DVT, VTE, PTE, complicated gangrene syndromes such as SPG, limb gangrene and acrocyanosis, which pathogeneses are summarized and thrombotic syndromes are exemplified in the bottom section of the Figure even though all of these syndromes have been rarely but previously described in other sepsis as well. This author does not think these thrombotic syndromes are the unique feature of COVID-19 but more likely have occurred as the result of intensive intervention utilizing numerous vascular devices and accesses in ICU following the directive from hyped psychology of political and medical communities in pandemic and also were highlighted by enthusiastic publications of the cases in medical literature. To emphasize the importance of vascular damage contributing to the heightened role inducing thrombogenesis in ICU setting, I have summarized the pathogenesis of several thrombotic syndromes that have occurred in COVID-19 infection. This Figure can be easily applied to other sepsis in place of COVID-19 sepsis. Abbreviations: ANF, acute necrotizing fasciitis; ARDS, acute respiratory distress syndrome; BCS, Budd-Chiari syndrome; CVST, cerebral venous sinus thrombosis; DVT, deep venous thrombosis; ECs, endothelial cells; EVT, extravascular tissue; ICU, intensive care unit; ITP, immune thrombocytopenic purpura; IVCT/SVCT, inferior vena cava thrombosis/superior vena cava thrombosis; MAHA, microangiopathic hemolytic anemia; MODS, multiorgan dysfuncion syndrome; MMT, micromacrothrombosis; MVT, mesenteric vein thrombosis; PTE, pulmonary thromboembolism; PVT, portal vein thrombosis; SET, subendothelial tissue; SPG, symmetrical peripheral gangrene; SVT, splanchnic vein thrombosis; TF, tissue factor; TTP, thrombotic thrombocytopenic purpura; ULVWF, ultra large von Willebrand factor; aEA-VMTD, arterial endotheliopathy-asasociated vascular microthrombotic disdase; vEA-VMTD, venous EA-VMTD; VCCS, venous circulatory congestion syndrome; VTE, venous throm-boembolism.

**Table 1 biomedicines-10-02706-t001:** Conceptual characteristics of the terms: hemostasis, coagulation and thrombosis.

	Hemostasis	Coagulation	Thrombosis ^+^
Term concept	Philosophical	Physiological	Structural
Implied meaning	Natural process in vivo	Artificial process in vitro	Pathological process in vivo
		Physiologic process at bleeding site	
Involved location	Blood vessel wall	Laboratory test tube	Intravascular lumen
		Vessel wall in extravascular trauma	
End products	Hemostatic plug	Fibrin mesh/fibrin clot *	Microthrombi/macrothrombus
Critical role in	Vascular wall injury	Coagulation test	Thrombogenesis
		Hemorrhage	
Involved components	Endothelium	TF/thromboplastin	ULVWF/FVIII from ECs
	SET	Coagulation factors	Platelets from circulation
	EVT	Fibrinogen	Coagulation proteins/serine proteases
	Blood from circulation		Fibrinogen
			TF from SET/EVT
End results	Determined by the depth	Determined by participating	Determined by ULVWF/FVIII, platelets, TF,
	of vascular damage	coagulation factors	hemostatic factors and unifying mechanism
Inciting events	Endotheliopathy	Coagulation tests for PT and aPTT	Microthrombosis (i.e., EA-VMTD)
	Vascular injury		Macrothrombosis (e.g., DVT)
			Combined micro-macrothrombosis (e.g., VTE)

Abbreviation: “DIC”, false disseminated intravascular coagulation; DVT, deep vein thrombosis; ECs, endothelial cells; EVT, extravascular tissue; EA-VMTD, endotheliopathy-associated vascular microthrombotic disease; MODS, multiorgan dysfunction syndrome; PT, prothrombin time; aPTT, activated partial thromboplastin time; SET, subendothelial tissue; TF, tissue factor; TTP, thrombotic thrombocytopenic purpura; ULVWF, ultra large von Willebrand factor multimers; VMTD, vascular microthrombotic disease; VTE, venous thromboembolism. * Only coagulation disorder in vivo is acute promyelocytic leukemia because fibrin clots are formed without platelets and causes hemorrhagic disease. ^+^ Thrombosis means platelets are involved.

**Table 2 biomedicines-10-02706-t002:** Three fundamentals in normal and abnormal hemostasis.

**(1) Hemostatic principles**		
(1) Hemostasis can be activated only by vascular injury.		
(2) Hemostasis must be activated through ULVWF path and/or TF path.		
(3) Hemostasis is the same process in both hemorrhage and thrombosis.		
(4) Hemostasis is the same process in both arterial thrombosis and venous thrombosis.		
(5) Level of vascular damage (endothelium/SET/EVT) determines different clinical phenotypes of hemorrhage and thrombosis.		
**(2) Major participating components**		
**Components**	**Origin**	**Mechanism**
(1) ECs/SET/EVT	Blood vessel wall/EVT	Protective barrier
(2) ULVWF/FVIII	ECs	Endothelial exocytosis/anchoring and microthrombogenesis
(3) Platelets	Circulation	Adhesion to ULVWF to form microthrombi/assembling and microthrombogenesis
(4) TF	SET and EVT	Release from tissue due to vascular injury/leading to fibrinogenesis
(5) Coagulation factors	Circulation	Activation to fibrin mesh and fibrin clot/participating in fibrinogenesis
**(3) Vascular injury and hemostatic phenotypes**		
**Injury-induced damage**	**Involved hemostatic path**	**Level of Vascular Injury and examples**
(1) Endothelium	ULVWF	Level 1 damage—microthrombosis (e.g., TIA [focal]; Heyde’s syndrome [local]; EA-VMTD [disseminated])
(2) Endothelium/SET	ULVWF + sTF	Level 2 damage—macrothrombosis (e.g., AIS; DVT; PTE; AA)
(3) Endothelium/SET/EVT	ULVWF + eTF	Level 3 damage—macrothrombosis with hemorrhage (e.g., THS; THMI)
(4) EVT alone	eTF	Level e damage—fibrin clot disease (e.g., AHS [e.g., SDH; EDH]; ICH)
**Hemostatic phenotypes**	**Causes**	**Genesis**
(1) Hemorrhage	External bodily injury	Trauma-induced external bleeding (e.g., accident; assault; self-inflicted)
(2) Hematoma	Internal EVT injury	Obtuse trauma-induced bleeding (e.g., tissue and cavitary hematoma)
(3) Thrombosis	Intravascular injury	Intravascular injury (e.g., atherosclerosis; sepsis; indwelling catheter; surgery)

Abbreviations: AA, aortic aneurysm; AHS, acute hemorrhagic stroke; AIS, acute ischemic stroke; DVT, deep venous thrombosis; ECs, endothelial cells; EDH, epidural hematoma; EVT, extravascular tissue; ICH, intracerebral hemorrhage; PTE, pulmonary thromboembolism; SDH, subdural hematoma; SET, subendothelial tissue; TF, tissue factor; eTF, extravascular TF; sTF, subendothelial TF; ULVWF, ultra large von Willebrand factor; THS, thrombo-hemorrhagic stroke; THMI, thrombo-hemorrhagic myocardial infarction, TIA, transient ischemic attack.

**Table 3 biomedicines-10-02706-t003:** Vascular system contributing to the phenotype expression of thrombosis.

**The depth of intravascular wall injury**
ECs injury: ULVWF and FVIII release (e.g., sepsis causing EA-VMTD)
ECs and SET injury: ULVWF and FVIII and TF release (e.g., vascular trauma causing DVT)
ECs and SET/EVT injury: ULVWF and FVIII and TF (e.g., vascular causing THS)
**The extent of injury affecting vascular tree system**
Focal/local/multifocal (e.g., TIA, DVT, Susac syndrome)
Regional (e.g., vascular trauma, Kasabach-Merritt syndrome)
Disseminated (e.g., endothelial damage/dysfunction due to sepsis, vaccination, envenomation and others)
**The vascular milieux system**
Venous (e.g., ITP-like syndrome, VOD, DVT, IVCT/SVCT, VTE, PTE, CVST)
Arterial (e.g., TTP-like syndrome, arterial thrombosis, AMI, AIS, SPG, diabetic gangrene)
Microvasculature (e.g., capillaries, arterioles, venules, and hepatic sinusoids, causing microthrombosis such as ARDS, encephalopathy, ALF, HUS, WFS)
Macrovasculature (e.g., aorta, artery and “minute” arteries, and vena cava, and vein, causing macrothrombosis such as aortic aneurysm, AIS, AMI, SPG, DVT, IVC/SVC, CVST)
**The locality of vascular injury**
Tropism (e.g., bacteria, virus, fungus, parasite causing organotropism such as WFS syndrome due to N. menningococcus)
Endothelial heterogeneity (e.g., gene expression of host in specific vascular system such as ITP-like syndrome in venous system)
Trauma site
**Interaction with non-hemostatic factors**
Environmental factor (e.g., pathogen, vaccine, venom, toxin, drug, chemical, trauma)
Hemostasis altering genetic factor (e.g., thrombophilic genes such as PC, PS, FV-Leiden, antithrombin, ADAMTS13, and others)
Hereditary disease (e.g., Fabry’s disease, HERNS syndrome, Degos disease, hereditary hemorrhagic telangiectasia)

Abbreviations: AIS, acute ischemic stroke, ALF, acute liver failure; AMI, acute myocardial infarction; ARDS, acute respiratory distress syndrome; CNS, central venous system; CVST, cerebral venous sinus thrombosis; DVT, deep venous thrombosis; EA-VMTD, endotheliopathy-associated vascular microthrombotic disease; ECs, endothelial cells; EVT, extravascular tissue; HERNS, hereditary endotheliopathy with retinopathy, nephropathy and stroke; HUS, hemolytic-uremic syndrome; IVC/SVC, inferior vena cava/superior vena cava; PC, protein C; PS, protein S; PTE, pulmonary thromboembolism; SET, subendothelial tissue; SPG, symmetrical peripheral gangrene; THS, thrombohemorrhagic stroke; TF, tissue factor; TIA, transient ischemic attrack; ULVWF, ultra large von Willebrand factor; VOD, veno-occlusive disease; VTE, venous thromboembolism; WFS, Waterhouse-Friderichsen syndrome.

**Table 4 biomedicines-10-02706-t004:** Endotheliopathy in arterial and venous milieux determining the clinical phenotype of thrombotic disorders.

Clinical Phenotype	Arterial Endotheliopathy	Venous Endothelipathy
**Underlying pathology**	aEA-VMTD	vEA-VMTD
**Physiological/hemodynamic difference in vascular milieu**	Efferent circulation from the heart (O_2_ delivery)	Afferent circulation into the heart (CO_2_ disposal)
	Tissue hypoxia (e.g., microthromboangiitis obliterans)	Pulmonary circulatory congestion (e.g., ARDS)
	High pressure flow	Low pressure flow
	High shear stress	Low shear stress
	Capillary and arteriolar microvascular event	Venous and pulmonary microvascular event
	Insignificant role of NETosis	Significant role of NETosis
**Primary cause**		
Vascular injury (ECs)	Sepsis-induced arterial microvascular endotheliopathy	Sepsis-induced venous endotheliopathy
		Vaccine-induced venous endotheliopathy
Vascular pathology site	Disseminated aEA-VMTD at microvasculature	Transient or “silent” vEA-VMTD at venous system
Activated hemostatic path	ULVWF path	ULVWF path
Thrombosis component	Microthrombi strings in the microvasculature	Microthrombi strings in venous system
Microthrombotic event	Disseminated VMTD	Silent microthrombosis with microthrombolysis
**Clinical phenotypes**	TTP-like syndrome	ITP-like syndrome
	consumptive thrombocytopenia	consumptive thrombocytopenia
	MAHA	MAHA (mild and rare: e.g., ARDS)
	MODS/MOIS	MOIS

Abbreviations: aEA-VMTD, arterial endotheliopathy-associated vascular microthrombotic disease; vEA-VMTD, venous-VMTD; ARDS, acute respiratory distress syndrome; ECs, endothelial cells; ITP, immune thrombocytopenic purpura; MAHA, microangiopathic hemolytic anemia; MODS, multiorgan dysfunction syndrome; MOIS, multiorgan inflammatory syndrome; NETs, neutrophil extracellular traps; TTP, thrombotic thrombocytopenic purpura; ULVWF, ultra large von Willebrand factor.

**Table 5 biomedicines-10-02706-t005:** Classification of primary and secondary phenotype examples of the thrombotic disorder.

**Microthrombosis**
**● aEA-VMTD**
**Mechanism:** via microthrombogenesis from activated ULVWF path due to ECs injury of capillaries and arterioles
Disseminated aEA-VMTD (e.g., TTP-like syndrome, MODS, MVMI, microthrombotic encephalopathy, HUS)
Regional aEA-VMTD (e.g., Kasabach-Merritt syndrome, Heyde’s syndrome)
Multifocal aEA-VMTD (e.g., HERNS syndrome, Susac syndrome, diabetic retinal microaneurysm)
Focal aEA-VMTD (e.g., TIA)
**● vEA-VMTD**
**Mechanism:** via microthrombogenesis from activated ULVWF path due to venous ECs injury
Disseminated vEA-VMTD (e.g., “silent” ITP-like syndrome, ARDS)
Regional vEA-VMTD (e.g., VOD)
**Macrothrombosis**
**● Arterial macrothrombosis**
**Mechanism:** via macrothrombogenesis from activated ULVWF path and TF path due to arterial ECs and SET/EVT injury
Localized arterial thrombosis (e.g., AIS, THS, AMI, aortic aneurysm-associated thrombosis)
Multifocal partially obstructive macrothrombosis (e.g., chronic atherosclerosis?)
**● Venous macrothrombosis**
**Mechanism:** via macrothrombogenesis from activated ULVWF path and TF path due to venous ECs and SET/EVT injury
Localized venous thrombosis (e.g., distal DVT, superficial venous thrombosis)
** Fibrin clot disease**
**● Arterial/venous fibrin clot disease**
**Mechanism:** via pathologic fibrinogenesis from activated aberrant TF path due to overexpressed TF in APL
Disseminated fibrin clot disease and hemorrhagic syndrome (e.g., true DIC in APL)
Localized hematoma (?) (e.g., internal bleeding including SDH, EDH, AHS)
** Combined micro-macrothrombosis (due to two different vascular injuries: e.g., sepsis and vascular injury in ICU )**
**● Arterial combined micro-macrothrombosis**
**Mechanism:** via combined micro-macrothrombogenesis from activated ULVWF path and TF path in arterial system
Peripheral gangrene syndrome (e.g., SPG, PDIS, Buerger’s disease, Fournier’s disease, purpura fulminans, acrocyanosis)
**● Venous combined micro-macrothrombosis**
**Mechanism:** via combined micro-macrothrombogenesis from activated ULVWF path and TF path in venous system
Venous circulatory congestion syndrome (e.g., VTE, PTE, IVCT/SVCT, SVT, BCS, PVT, CVST)
**Non-hemostatic thrombotic syndromes**
**● Disseminated VMTD**
**Mechanism:** excess of ULVWF due to ADAMTS13 deficiency leading to intravascular microthrombogenesis
ADAMTS13 gene mutation-associated vascular microthrombotic disease (GA-VMTD: Hereditary TTP)
ADAMTS13 gene antibody-associated vascular microthrombotic disease (AA-VMTD: Acquired TTP)
**● HIT-WCS**
**Mechanism:** platelet thrombi due to heparin-PF4 complex reacting with heparin-PF4 complex antibody
White clot syndrome
**● Fibrin clot disease in APL ***
**Mechanism:** activated aberrant TF path from overexpressed TF of APL without vascular injury
APL coagulopathy with hemorrhagic syndrome (i.e., fibrin clot disease; true DIC)

Abbreviations: AHS, acute hemorrhagic stroke; AIS, acute ischemic stroke; AMI, acute myocardial infarction; APL, acute promyelocytic leukemia; ARDS, acute respiratory distress syndrome; BCS, Budd-Chiari syndrome; CVST, cerebral venous sinus thrombosis; DVT, deep venous thrombosis; AA-VMTD, antibody-associated vascular microthrombotic disease; aEA-VMTD, arterial endotheliopathy-associated VMTD; vEA-VMTD, venous EA-VMTD; GA-VMTD, gene mutation-associated VMTD; DIC, disseminated intravascular coagulation; ECs, endothelial cells; EDH, epidural hematoma; EVT, extravascular tissue; HERNS, hereditary endotheliopathy, retinopathy, nephropathy and stroke; HIT-WCS, heparin-induced thrombocytopenia with white clot syndrome; HUS, hemolytic-uremic syndrome; ITP, immune thrombocytopenic purpura; IVCT/SVCT, inferior vena cava thrombosis/superior vena cava thrombosis; MODS, multiorgan dysfunction syndrome, MVMI, microvascular myocardial infarction; PDIS, peripheral digit ischemic syndrome; PNH, paroxysmal nocturnal hemoglobinuria; PTE, pulmonary thromboembolism; PVT, portal vein thrombosis; SDH, subdural hematoma; SET, subendothelial tissue; SPG, symmetrical peripheral gangrene; SVT, splanchnic vein thrombosis; THS, thrombo-hemorrhagic stroke; TF, tissue factor; TIA, transient ischemic attack; TTP, thrombotic thrombocytopenic purpura; ULVWF, ultra large von Willebrand factor; VMTD, vascular microthrombotic disease; VOD, veno-occlusive disease; VTE, venous thromboembolism. * This condition may be classified as thrombosis in either hemostatic column or non-hemostatic column for educational purpose.

**Table 6 biomedicines-10-02706-t006:** Expected laboratory and clinical markers of different phenotypes of thrombosis based on two hemostatic theories in vivo.

	aEA-VMTD	vEA-VMTD	aMT	vMT	FCD	Combined aMMT	Combined vMMT
**Typical example**	TTP-like syndrome	ITP-like syndrome	AIS	Distal DVT	APL	SPG	VTE
**Thrombocytopenia**	+	+	-	-	+ (due to leukemia)	+	+
**Fibrinogen**	Increased/decreased	Increased/decreased	Normal	Normal	Decreased	Markedly increased	May be increased
**ULVWF/VWF antigen**	Markedly increased	May be increased	Normal	Normal	Normal expected	Markedly increased	Maybe increased
**FVIII**	Markedly increased	Increased	Normal	Normal	Markedly decreased	Increased	May be increased
**FV**	Normal	Normal	Normal	Normal	Decreased	Decreased (?)	Decreased (?)
**ADAMTS13**	Low	Low	Normal	Normal	Normal expected	Likely low	Likely low
**FSPs**	+/-	+/-	Normal	Normal	Positive	Positive	Positive
**D-dimer**	Normal	Normal	+/-	+/-	Increased	Markedly increased	Markedly increased
**MAHA**	+	+/-	-	-	-	+	-
**MODS**	+	+/-	-	-	-	+	-
**Inflammation**	+	+	-	-	?	+	+
**C5b-9 involvement**	Expected to be +	Expected to be +	-	-	?	Expected to be +	Expected to be +

Abbreviations: AIS, acute ischemic stroke; APL, acute promyelocytic leukemia; aEA-VMTD, arterial endotheliopathy-associated vascular microthrombotic disease; vEA-VMTD, venous EA-VMTD; FCD, fibrin clot disease; ITP, immune thrombocytopenic purpura; MAHA, microangiopathic hemolytic anemia; MODS, multiorgan dysfunction syndrome; aMT, arterial macrothrombosis; vMT, venous macrothrombosis; aMMT, arterial micro-macrothrombosis; vMMT, venous MMT; TTP, thrombotic thrombocytopenic purpura; SPG, symmetrical peripheral gangrene; VTE, venous thromboembolism; ULVWF, ultra large von Willebrand factor; VWF, von Willebrand factor.

## Data Availability

Data supporting the medical theories are in public dormains. Application of the paper is derived from public data and the hypothesis is from author’s discovered theories.

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
