# Peer review of "Novel Classification of Thrombotic Disorders Based on Molecular Hemostasis and Thrombogenesis Producing Primary and Secondary Phenotypes of Thrombosis"

_biomedicines, 2022, doi:10.3390/biomedicines10112706_

Round 1

Reviewer 1 Report

The manuscript describes an interesting review of  about novel classification of thrombotic disorders. Hemostasis, endowed to human to protect lives, is a process of logical blood clotting system to prevent blood loss in vascular injury. However, the notion that deadly thrombosis occurs as a result of normal hemostasis in intravascular injury could encounter with conceptual skepticism because the term ‘thrombosis’ automatically conjures up as serious disease.

I would like to emphasize that this is a well-crafted review. However, there are a few things that need to be corrected.

Page 5 lines 113-120. The authors should make an important claim about fibrinogen . ,,Fibrinogen contributes to various pathological events including thrombosis due to decrease of plasma concentration of fibrinogen, its changed structural properties, or from the effect of polymorphisms on clot stiffness, permeability, and resistance to lysis.“ This statement was published in a manuscript that should be cited: Brunclikova et al., J. Clin. Med. 2022, 11(4), 1083; https://doi.org/10.3390/jcm11041083

Pages 9-10 lines 630-655. The authors describe fibrin clot disease. ,,In absence of fibrinogen or at low levels, the small amount of thrombin usually formed remains longer in the circulation as no or less sequestering on circulating fibrinogen occurs (i.e. antithrombin function of fibrin is impaired). Besides, thrombin generation has been shown to be increased in the plasma of patients with low levels of fibrinogen. Such traces of circulating thrombin may precipitate a fraction of fibrinogen in infusion. In hypofibrinogenemia, usually only normal fibrinogen variant reaches the circulation and therefore the fibrin clot function is not altered“. The this statement should be added to the manuscript. These facts was published in a manuscript which should be cited by Simurda T et al. Thromb Res. 2020 Apr;188:1-4. doi: 10.1016/j.thromres.2020.01.024.

The tables and figures  are processed correctly.

I have to say that with these 87 references,  most references are newer than 5 years old.

Author Response

Author’s Response to Reviewer 1

1)   On general comments.

Thank you for your positive comments on my proposed “novel classification of the thrombotic disorders” and craftmanship of my works.

2)   Page 5 lines 113-120 in legend of Table 2 on the role fibrinogen

Yes, I have shown the role of fibrinogen” via fibrinogenesis in the pathway of TF path producing fibrin clots, which is shown from TF  to “fibrinogen” and “fibrin”, eventually leading to  macrothrombogenesis (Figure 1). Following the activation of TF-FVIIa complex leading to coagulation cascade, vascular injury produces thrombin, which convert fibrinogen to fibrin. Therefore, fibrinogen is a critical component in fibrinogenesis. This coagulation cascade is called TF pathway, which is well known to clinicians as I have described in page 5 and text lines 138-140 (I have noted “converts prothrombin to thrombin and facilitates fibrinogen to fibrin clot formation”). To focus on thrombogenesis, I have not elaborated already well-known TF pathway in detail because this paper is to emphasize the newer role of “ULVWF” path as well as TF (fibrinogenetic) path.

3)   Page 9-10 lines 630-655 on fibrin clot disease.

Your comments on thrombin generation and its interaction with fibrinogen is interesting. I have not had in-depth knowledge of thrombin dynamics in afibrinogenemia and hypofibrinogenemia in circulation. This manuscript is primarily to look into thrombogenesis producing “thrombosis” caused by a vascular injury. Fibrin clot disease only occurs in APL without a vascular damage and is not true “thrombotic disorder” although it can be called true “DIC” as I have mentioned in the text causing hypofibrinogenemia and consumption of FVIII and FV. Additionally, the pathogenesis of fibrin clot disease in APL is yet to be confirmed because of its aberrant nature of hemostasis. Thus, fibrin clot disease should not be fitted into “thrombotic disorders” as I have mentioned in Table 5, and I am no qualified to explain a specific functional nature on fibrinogen.

4)   The Tables and Figures are processed correctly.

Thank you for your comment. However, per request of academic editor, I have further clarified them for easy comprehension for all 6 Tables.

5)   In regard to 87 references with relatively recent publications.

Classification of the thrombotic disorders based on molecular hemostasis has not been not possible to date because we have not had complete hemostatic mechanism in vivo, and thrombogenesis has been essentially unknown beyond Virchow’s triad more than 150 years. Now we have good concept.

6)   Thank you very much for critiquing and contributing to my classification of thrombotic disorders.

Reviewer 2 Report

This review is about thrombotic disorders, the author suggested the new classification of the disorders. I wrote my comments below.

Although the author classified the thrombotic disorders based on some conditions, it is very difficult to separate these disorders clearly. Please add the limitation that these disorders cannot be categorized  clearly.

In tables, there are too much information and words. Please make clear and reconsider the composition.

Author Response

Author’s response to Reviewer 2

1)    I have utilized three mechanisms of thrombogenesis, which include 1) microthrombogenesis forming “microthrombi strings” made of platelet-ULVWF complexes, 2) fibrinogenesis forming “fibrin meshes/clots”, 3) macrothrombogenesis producing “macrothrombous” from unifying mechanism of microthrombi and fibrin meshes. These mechanisms clearly explain every thrombotic disorder as shown in Table 5, which I have simplified for better comprehension as requested by Academic Editor, Reviewer 1 and you now.

2)   There are two unresolved issues in thrombogenetic mechanisms. One is how fibrinogenesis interact with factor VIII in hemostasis in vivo to form macrothrombosis to explain laboratory tests of intrinsic and extrinsic cascade mechanisms that have been tested by aPTT and PT and hemostatic theory in vivo. The other is how unifying molecular mechanism of microthrombi strings and fibrin meshes occur. I am researching these issues now. I hope to publish them in near future. However, I consider the classification of the phenotypes of the thrombotic disorder is solidly established. Please see Table 5, Figure 5 and Figure 6. For example, Figure 6 logically explains every thrombotic disorder encountered in COVID-19 by utilizing “two-activation theory of the endothelium” and “two-path unifying theory” of hemostasis.

3)    Tables are revised as you and academic editor have requested. Every word is carefully selected, but it is true that they contain complex components but are necessary contents.

4)   I sincerely appreciate your time efforts helping for the progress of science. Many thanks.

Reviewer 3 Report

In the provided manuscript, Chang proposed novel classification of thrombotic disorders based on five logic criteria, in a way to unify widely accepted “two-activation theory of the endothelium” and “two-path unifiyng theory” of hemostasis. These are presented in Introduction part of the review. In a search of new classification of thrombotic disorders author considered vascular, environmental, and genetic components contributing to the phenotype expression. Finally, author proposed four primary thrombotic phenotypes produced by different thrombogeneses: (1) microthrombosis, (2) macrothrombosis, (3) fibrin clot disease, and (4) combined micro-macrothrombosis. Author also took efforts to consider the accuracy of principal terms, such as “thrombosis”, “blood clots” or “hemostasis” and their descriptive function using quite colorful language.

The idea to implement novel classificaltion of thrombotic disorders is vital since there is close connection between clinical (on even common) description of thrombotic phenotype and clinical/pharmacological treatment. Hence, open and wide discussion upon novel classification of thrombotic disorders is clinically and cognitively needed.

Concerns:

-         - In case of macrothrombosis (point 6.2.2), I have a problem to accept that “This pathogenesis is simple and straight forward and almost always caused by a local vascular injury and tend to be localized without activation of inflammatory pathway.“ (lines 627-629) since triggers to initiate and propagate fibrin formation are different in case of arterial thrombosis (which is directly associated with platelet procoagulant resposne) and in case of venous thrombosis. This is expressed in different morphology and quantitative ratio of cellular elements (RBC, platelets etc.) and fibrin (types) in arterial versus venous clots [Chernysh et al. The distinctive structure and composition of arterial and venous thrombi and pulmonary emboli. Sci Rep. 2020; 10: 5112] 

-        Engagement of platelets in thrombogenesis (including thrombin generation -> fibrin formation) should be discussed with special attention. Platelet secretion of coagulation factors and local procoagulant response associated with phosphatidylserine exposure are expected to be vital events in the main context. Antiplatelet pharmacotherapy should be also mentioned in appropriate sections.

-          There are apparently missing references in numerous paragraphs, e.g. lines 608, 615, 620, 739, etc.

-          Did author consider to separate the COVID 19 section from the whole test body as independent manuscript?

Author Response

Author’s response to Reviewer 3

1)   I have enjoyed debating with scientists of hemostasis via this forum of critiques on submitted manuscripts in the past because it is mutually learning experience. I have appreciated reading your critique.

2)   In a search of new classification of thrombotic disorders.

It is important to understand “thrombosis’ is initiated only by “vascular injury” alone since without a vascular injury there is no release of ULVWF/FVIII and/or TF because they are the essential components promoting thrombogenesis via activated ULVWF path and/or TF path. In another word, there is no hemostasis is needed and occurs without these elements. Therefore, underlying thrombophilic states such as protein C and protein S, hemophilic states such as hemophilic genes A and B, environmental factors such as stasis, smoking, Shiga toxin and spike protein of SARS-CoV-2 are influencing factors only after vascular injury is done. This is the most important principle in hemostasis as affirmed in Table 2.

3)   Recognizing the clinical phenotypes are essential in the selecting pharmacological decision making for the individualized treatment.

Yes, you are absolutely correct. Without the understanding of the pathogenesis for each specific phenotype, random selection clinical trials utilizing old and new drugs to find “evidence-based -medicine” are wasteful, time-consuming and unproductive. For example, EA-VMTD in sepsis (e.g., ARDS) should never be treated with anticoagulant such as TF inhibitors heparin or coumadin, but should be treated with ULVWF inhibitors such as rADAMTS13 or antic-complement agent [14,21]. No wonder is why clinical and coagulation communities have failed with numerous clinical trials without comprehension of pathogenesis of septic endotheliopathy in the past.

4)   Concern 1: Local nature of “macrothrombosis” in a local vascular injury

The concept of septic endotheliopathy is easily understood if clinician accept it causes disseminated endothelial (damage limited to ECs) vascular disease leading to “disseminated VMTD” via ULVWF path without bleeding. On the other hand, a local bleeding trauma causes ECs +SET+ EVT damage, which causes “local macrothrombosis” via local ULVWF path and TF path. Thus, VMTD is much more serious disease and local macrothrombosis is a transient disease. Yes, it is true microthrombosis does not contain NETs, but macrothrombosis, especially venous one compared to arterial one, contains significant components of NETs because venous circulation is slow and congested and NETs are “passively integrated”, much more so in VTE than solitary DVT, and compared to arterial thrombosis (arterial macrothrombosis) and arterial gangrene (arterial combined micro-macrothrombosis). That is the reason why gangrene syndrome and VTE become much more serious arterial or venous combined micro-macrothrombosis respectively.

5)   Concern 2: Different morphology and cellular contents in arterial vs. venous blood clots.

You are right. These are important factors which have not been discussed much in the thrombosis literature to date. I call it difference in “vascular milieux” on the thrombogenesis. Examples are venous vs. arterial, microvasculature vs, macrovasculature, sinusoid vs, capillary etc. Certainly, these difference result in different characters of blood clots (e.g., macrothrombi vs. macrothrombi, vs fibrin clots) as well as different amounts of NETS, different phenotypes (e.g., TTP-like syndrome vs. ITP-like syndrome, VTE vs. SPG) and mechanism of pulmonary emboli in venous EA-VMTD [10,11] and summarized Table 4 and Table 5.

I am very grateful to you for your review of my manuscript and constructive critique, and am happy to learn your special interest in the mechanism of hemostasis in vivo.

6)   Engagement of platelet in hemostasis

Platelets are utilized mostly in microthrombogenesis to form microthrombi strings and produce “consumptive thrombocytopenia” in hemostasis in vivo. However, eventually in macrothrombogenesis, the platelet in microthrombi strings and thrombin in fibrin meshes come together in one part of unifying mechanism. Indeed, platelets are not present in fibrin meshes until they meet together in unifying mechanism. Please recognize that platelets are called “thrombocytes”. When thrombosis contains platelets in their character of blood clots, it is correctly designated as “thrombotic disorders”, but fibrin clot disease is not called as thrombotic disorder because it does not contain platelets. Also, in coagulation laboratory, the tests of aPTT, PT, thrombin time, and clotting time are called “coagulation tests”, but not called “thrombosis tests” because platelets are not included in the laboratory test tube. It is also proof that laboratory coagulation tests do not represent the process what is happing in hemostasis in vivo/thrombogenesis. 

I do not have good information on platelet secretion of coagulation factors and local protein-induced coagulation response. I will try to familiarize with it, but is beyond of the discussion of thrombogenesis at this time. As shown, in “two-activation theory of the endothelium” and “two-path unifying theory” of hemostasis, the therapeutic benefit of antiplatelet agent is limited against microthrombogenesis, which suggests antiplatelet therapy may have a significant benefit for prevention (e.g., TIA), but expected to have little significant benefit for formed macrothrombosis (e,g., DVT and arterial thrombosis), but no benefit at all for combined micro-macerothrombosis (e.g., VTE and gangrene syndrome). Therefore, I have not included the therapeutic pitfalls in this classification article.

7)   Missing references.

They are not missing reference parentheses, but in the first draft production Editorial office apparently automatically deleted every Table and Figure and their numbers in all of the text. While in proof-reading process, I have reinserted their correct captions in all () spaces.

I sincerely express my gratitude reviewing this manuscript. You are one of scientists in frontline of medicine determining our future in medical community.

Round 2

Reviewer 2 Report

I confirmed the revised manuscript. I can accept.

Reviewer 3 Report

I appreciate author's response and I feel satisfied. The manuscript is worth to be published and discussed by the community. Future expansion of this review, including more deeply considered role of platelets are awaited.